# Plasmalogen Deficiency and Overactive Fatty Acid Elongation Biomarkers in Serum of Breast Cancer Patients Pre- and Post-Surgery—New Insights on Diagnosis, Risk Assessment, and Disease Mechanisms

**DOI:** 10.3390/cancers13164170

**Published:** 2021-08-19

**Authors:** Satomi Tomida, Dayan B. Goodenowe, Teruhide Koyama, Etsuko Ozaki, Nagato Kuriyama, Midori Morita, Yasuyo Yamazaki, Koichi Sakaguchi, Ritei Uehara, Tetsuya Taguchi

**Affiliations:** 1Department of Endocrine and Breast Surgery, Kyoto Prefectural University of Medicine, Kyoto 602-8566, Japan; satomida@koto.kpu-m.ac.jp (S.T.); midori@koto.kpu-m.ac.jp (M.M.); ksak@koto.kpu-m.ac.jp (K.S.); ttaguchi@koto.kpu-m.ac.jp (T.T.); 2Department of Epidemiology for Community Health and Medicine, Kyoto Prefectural University of Medicine, Kyoto 602-8566, Japan; ozaki@koto.kpu-m.ac.jp (E.O.); nkuriyama@s-sph.ac.jp (N.K.); ruehara@koto.kpu-m.ac.jp (R.U.); 3Prodrome Sciences USA LLC, Temecula, CA 92591, USA; d.goodenowe@prodromesciences.com (D.B.G.); Y.Yamazaki@prodromesciences.com (Y.Y.); 4Shizuoka Graduate University of Public Health, 420-0881 Shizuoka, Japan

**Keywords:** breast cancer, fatty acid, ELOVL5, plasmalogen

## Abstract

**Simple Summary:**

Breast cancer (BC) is the most commonly diagnosed cancer in women. Mammography and ultrasonography are commonly used for BC screening; however, they are associated with problems such as inconvenience, radiation exposure, and dependence on the skill level of operators. To overcome this problem, we performed a comprehensive lipid metabolomic analysis of serum using high-resolution accurate mass spectrometry from two case-control studies that included non-BC, BC subjects pre-surgery and BC subjects one-month post-surgery to determine if the metabolic signatures of over-active fatty acid elongation and other lipid changes could be detected in BC vs. non-BC subjects. The ratios of the linoleic acid to the oleic acid which were evaluated in multiple lipid pools were lower in pre-surgery BC subjects, however, these ratios increased at post-surgery and were no longer different from non-BC subjects. On the other hand, the ethanolamine plasmalogen levels were lower in pre-surgery BC subjects and were not recovered by surgical removal. These do not appear to be caused by BC tumor activity and may be pre-existent and a possible risk factor for BC. In this study, we have identified several lipid metabolic systems that detect both BC risk and BC activity.

**Abstract:**

The polyunsaturated fatty acid (PUFA) elongase, ELOVL5, is upregulated in breast cancer (BC) vs. adjacent normal tissue. We performed a comprehensive lipid metabolomic analysis of serum using high-resolution accurate mass spectrometry from two case-control studies that included non-BC, BC subjects pre-surgery, and BC subjects one-month post-surgery to determine if the metabolic signatures of over-active fatty acid elongation and other lipid changes could be detected in BC vs. non-BC subjects: study 1 (*n* = 48: non-BC, *n* = 69: pre-surgery BC); study 2 (blinded validation: *n* = 121: non-BC, *n* = 62: pre-surgery BC, *n* = 31: one month post-surgery). The ratio of the ELOVL5 precursor, linoleic acid (18:2) to a non-ELOVL5 precursor, oleic acid (18:1) was evaluated in multiple lipid pools (phosphatidylethanolamine (PtdEtn), phosphatidylcholine (PtdCho), lyso-PtdCho, and free fatty acids). This ratio was lower in pre-surgery BC subjects in all pools in both studies (*p* < 0.001). At one-month post-surgery, the 18:2/18:1 ratios increased vs. pre-surgery and were no longer different from non-BC subjects (*p* > 0.05 expect for lyso-PtdCho). In contrast to the elongation biomarkers, docosahexaenoic acid (22:6n-3) containing ethanolamine plasmalogen (EtnPls) species were observed to be further decreased in BC subjects one-month post-surgery vs. pre-surgery levels (*p* < 0.001). These results are consistent with the hypothesis that ELOVL5 is upregulated in BC tissue, which would result in the selective depletion of 18:2 vs. 18:1 containing lipid species. Surgical removal of the tumor removes the overactive ELOVL5 effect on serum lipids. In contrast, the low EtnPls levels do not appear to be caused by BC tumor activity and may be pre-existent and a possible risk factor for BC. These results indicate that it may be possible to screen for both breast cancer risk and breast cancer activity using a simple blood test.

## 1. Introduction

Breast cancer (BC) has the highest cancer-related morbidity rate among Japanese women [1]. The five-year survival rate is more than 90% if BC is diagnosed at stage 1 or 2. Accordingly, early diagnosis results in a good prognosis. It is important to diagnose BC and initiate treatment as early as possible. However, the BC screening rate among Japanese people is lower than in other countries (approximately 40%) [2], and many Japanese women may miss the opportunity for early detection of BC. Mammography and ultrasonography are commonly used for BC screening; however, they are associated with problems such as inconvenience, radiation exposure, and dependence on the skill level of operators. A minimally invasive and convenient serum-based test that can accurately identify BC patients would increase the screening rate and result in detecting BC at an earlier stage.

Several epidemiological studies have described the role played by dietary factors in determining the risk of BC [3,4,5]. It has been speculated that fatty acids are the dietary component associated with the occurrence of BC [6,7,8,9]; however, this remains controversial. A meta-analysis of prospective cohort studies found no association between serum fatty acid levels and risk of breast cancer [7]. On the other hand, some experimental studies found a relationship between fatty acid levels and the risk of BC. The very long-chain fatty acid elongase (ELOVL) plays an important role in fatty acid elongation [10]. Mammals have 7 elongases (ELOVL1–7), each with different substrate specificities [10]. ELOVL5 elongates omega-3 (n-3) and omega-6 (n-6) polyunsaturated acids and prefers C18–C22. Yamashita et al. reported that the level of linoleic acid (18:2) was lower in BC tissues and suggested that ELOVL5 accelerated elongation [11]. The researchers also reported that the mRNA expressions of ELOVL1, 5, and 6 were significantly higher in BC tissues than in corresponding normal breast tissues [11].

We have previously described the low levels of 28 carbon polyhydroxylated and the polyunsaturated long-chain fatty acid (C_28_H_46_O_4_), named GTA-446 in colorectal cancer patients, and the low levels of 36 carbon polyhydroxylated and the polyunsaturated long-chain fatty acid (C_36_H_66_O_6_), named PTA-594 in pancreatic cancer patients [12,13,14,15]. We also described low levels of plasmalogens in both patients [12,13,14,15]. To identify the metabolic signatures of up-regulated fatty acid elongation and other lipid changes in BC vs. non-BC subjects, we performed a comprehensive lipid metabolomic analysis of serum using high-resolution accurate mass spectrometry in two case-control studies in completely different cohorts that included non-BC and BC pre-surgery subjects as well as BC subjects one month after surgery.

## 2. Results

### 2.1. Discovery Phase

Table 1 shows the characteristics of the enrolled BC patients and healthy controls. Comprehensive, non-targeted high-resolution mass spectrometry of the discovery phase (study 1) serum samples resulted in over 2000 unique sample-specific accurate masses. Accurate masses corresponding to specific species of free fatty acids (FFA), phosphatidylethanolamine (PtdEtn), phosphatidylcholine (PtdCho), lyso-PtdCho, and ethanolamine plasmalogens (EtnPls) were observed to have significantly different levels in BC vs. non-BC persons (Appendix A). Appendix A shows 18:2 (linoleic acid)/18:1 (oleic acid) ratios are significantly lower with the *p*-value < 0.0005–0.000001 between BC patients and healthy controls on the red plots; metabolites of FFA, lyso-PtdCho, PtdEtn, and PtdCho series. Based on this analysis, a subset of lipid metabolites of interest was then created using key representative species of each of these metabolite classes for subsequent analyses and validation.

### 2.2. Biochemical System Validation

All study 1 samples along with all study 2 samples, which were sent to the laboratory blinded, were then processed in a manner identical to that of the original discovery phase; however, only the pre-selected lipid metabolites of interest were investigated. The original lipid metabolomic system changes observed in the discovery phase were re-affirmed upon re-analysis, attesting to the reproducibility of the extraction and analysis procedures. After receiving the data from the laboratory, the identities of study 2 samples were revealed, and the results were interpreted with respect to study cohort, disease stage, and pre-surgery vs. post-surgery status.

### 2.3. Validation Cohort Analysis

The effect of BC on the lipid metabolomic system was observed to be similar in both the discovery and validation data sets (Appendix A). Figure 1 illustrates the reproducibility of the lower 18:2/18:1 ratio across multiple lipid pools in both study cohorts. To further illustrate and compare the selective depletion of 18:2 species and EtnPls in BC, representative pairs of PtdEtn and ethanolamine plasmalogens (PlsEtn) species were normalized to PtdEtn 18:0/18:1 (Figure 2 and Figure 3). For details about this normalization, see Materials and Methods. The level of PtdEtn 18:0/18:1 was unchanged in BC and control subjects; however, both the levels of PtdEtn 18:0/18:2 and EtnPls 18:0/18:2 species were decreased in BC subjects relative to control (Figure 2). Likewise, the levels of key polyunsaturated arachidonic acid (20:4n-6) and docosahexaenoic acid (22:6n-3) were decreased, and this decrease was more severe in the EtnPls lipid pool. Figure 3 illustrates the selective effect of surgical tumor removal on the restoration of 18:2 species in both the PtdEtn and EtnPls pools; however, the levels of polyunsaturated n-6 and n-3 species were either unaffected or further reduced in post-surgery. Figure 4 illustrates the reproducibility of the normalizing effect of surgery on the 18:2/18:1 ratio in multiple lipid pools. In addition to EtnPls species and 18:2 containing phospholipids, lyso-PtdCho species were also observed to be reproducibly decreased in BC vs. non-BC subjects (Figure 5). No effect of the cancer stage was observed for any of the biomarkers investigated. Figure 6 illustrates the lack of effect of the cancer stage on several key biomarkers described above.

Despite the consistency of the above group effects, variability existed within individual persons as to which lipid pools were most affected in BC. To determine the accuracy of the observed biochemical changes in diagnosing BC, we generated a multivariate model comprising the 18:2/18:1 ratio, total EtnPls, and total lyso-PtdCho. This approach resulted in excellent sensitivity and specificity as shown by an ROC curve with an AUC of 0.930 in study 2 as the validation phase (Figure 7). 

## 3. Discussion

In the present study, we used a non-targeted lipid metabolomics discovery platform to identify putative lipid metabolic systems of interest in BC. This approach has been successfully used previously to identify biomarkers that play roles in Alzheimer’s disease [12], autism [13], colorectal cancer [14,15], and pancreatic cancer [16,17]. Using this strategy, we identified several lipid metabolomic systems that were potentially altered in persons with BC. To eliminate false interpretations due to potential variations in analytical or clinical collection procedures, the original discovery serum samples were re-extracted and re-analyzed along with serum samples collected from a second case-control cohort recruited at a different time. We, therefore, can report with confidence that persons with BC have a lower 18:2/18:1 ratio, lower EtnPls levels, and lower lyso-PtdCho levels than persons without BC. Furthermore, of the biochemical systems we confirmed to be altered in persons with BC, only the 18:2/18:1 ratios returned to normal following surgical tumor removal. All other observed changes either remained the same or worsened. These observations indicate that two overlapping biochemical dysfunctions occur in BC. One type, exemplified by low EtnPls and lyso-PtdCho levels, predates tumor formation and may represent an increased susceptibility to BC. Another type, exemplified by the low 18:2/18:1 ratios, occurs only during tumor formation and growth.

We hypothesize that the observed selective depletion of 18:2-containing lipid species relative to 18:1-containing lipid species (Figure 1) occurs consequent to an upregulation of the polyunsaturated fatty acid elongase, ELOVL5. Fatty acid elongases play an important role in the synthesis of very long-chain fatty acid-containing membrane lipids from shorter chain dietary precursors. 20:4n-6 and 22:6n-3 containing membrane lipids are the main products of these reactions. Mammals have 7 elongases (ELOVL1–7) each of which has different substrate specificities [10]. ELOVL5 elongates n-3 and n-6 polyunsaturated acids and prefers C18–C22. A previous study has shown that the level of 18:2 was lower in BC tissues and was suggested to accelerate elongation due to ELOVL5. The mRNA expressions of ELOVL1, 5, and 6 were significantly higher in BC tissues than in the corresponding normal breast tissues [11]. Our results agree with these reports. In our study, the 18:2/18:1 ratios within multiple lipid classes were significantly decreased in BC patients prior to surgery. One-month postoperatively, these ratios were elevated compared with the preoperative ratios and were not significantly different from those of the healthy controls. These results indicate that the overactivity of ELOVL5 is limited to the BC tumor and that surgical tumor removal eliminates this effect of the overactivity on serum lipids. The levels of the products of n-6 and n-3 elongation and desaturation (20:4n-6 and 22:6n-3) were also observed to be decreased in BC (Figure 2). However, unlike the precursor 18:2, the levels of these products were not restored following surgical tumor removal (Figure 3). The lack of effect of tumor removal on systemic (blood) levels of 20:4n-6 and 22:6n-3, further supports our hypothesis that the excess demand of these metabolites was localized to the tumor.

The levels of PlsEtn containing 20:4n-6 and 22:6n-3 in BC subjects are consistently lower relative to the levels of PtdEtn containing 20:4n-6 and 22:6n-3 in BC subjects (Figure 3). EtnPls are manufactured in peroxisomes. Peroxisomal beta-oxidation generates acetyl-CoA. Within the peroxisome, this acetyl-CoA is used to synthesize the fatty alcohol that becomes the sn-1 vinyl ether of EtnPls. Peroxisomal acetyl-CoA is also the main source of cytosolic acetyl-CoA. Cytosolic acetyl-CoA is an obligate building block used for fatty acid elongation. In addition, peroxisomal beta-oxidation is the final step in the biosynthesis of 22:6n-3. Accordingly, impaired peroxisomal beta-oxidation would result in a simultaneous decrease in the levels of EtnPls, 22:6n-3, and 20:4n-6. A pre-existing state of impaired peroxisomal function as a risk factor for BC is consistent with these observations.

The physiological importance of EtnPls has been studied extensively [18]. The roles of EtnPls in mammalian physiology can be broadly grouped into two categories: membrane structure and function and antioxidation. EtnPls are essential for maintaining and regulating membrane cholesterol levels [19], membrane fusion [20], and virtually all membrane functions related to fluidity and fusion. Clinically, low serum EtnPls levels are observed in various cancers [21,22,23] and cardiovascular disease [24,25]. We previously confirmed that serum EtnPls levels were reduced in pancreatic cancer patients [17]. In the present study, we confirmed that total PLE levels were significantly decreased in the sera of BC patients and that they remained significantly lower than those among healthy controls postoperatively. Our study supports the hypothesis that EtnPls downregulation in the sera of BC patients is not derived from BC tissues but predates cancer activation.

Considering the mechanistic role of EtnPls in cholesterol regulation [19], particularly high-density lipoprotein (HDL)-mediated cholesterol efflux [26], the long-established role of abnormal intracellular cholesterol regulation in cancer [27,28,29], and the recent observation that inducing HDL-mediated cholesterol efflux yields an anti-proliferative and pro-apoptotic effect in BC cells [30], it is clear that reduced peroxisomal function, dysregulated intracellular lipid synthesis, and membrane structure defects all contribute to an increased risk of BC.

The histological examination of primary BC tissue [31] indicated that low AMP-activated protein kinase (AMPK) phosphorylation activity is a characteristic of approximately 90% of BC tumors and that underactive AMPK correlates with overactive acetyl-CoA carboxylase (ACC). The BC susceptibility gene 1 (*BRCA1*) encodes a protein that interacts and binds to the inactive, AMPK-phosphorylated form of ACC1 [32]. Accordingly, the biological function of BRCA1 is to maintain, extend, and regulate the suppression of ACC1 following its inactivation by AMPK. The BRCA1 protein found in women with *BRCA1* mutations is unable to perform this essential function. Collectively, these data suggest that sporadic and BRCA-mediated BC share a common metabolic path and that the presence of a BRCA mutation dramatically increases the risk of that path being activated.

The cell membrane phospholipid composition affects the protein composition and activity of the cell. We have previously shown that increasing membrane levels of 22:6n-3 EtnPls increases the levels of cholesterol esterifying enzymes [19] and non-amyloidogenic amyloid precursor processing enzymes [33]. Recently, it has been shown that 22:6n-3 supplementation increases the membrane 22:6n-3 levels in rats and that this increase resulted in a decreased incidence of experimentally induced BC [34]. It is of particular interest that the level of the BRCA1 protein was elevated in 22:6n-3-supplemented animals. Treatment of breast and ovarian cancer cells with 22:6n-3 has resulted in consistent positive effects [35,36,37]. Several studies have now shown that high dietary consumption of omega-3 fatty acids is associated with a decreased risk of BC [38].

Collectively, the results indicate that the low 22:6n-3 EtnPls level observed in BC could increase the susceptibility of cells to the overactivity of ACC by reducing the expression of BRCA proteins and reducing the ability of the cell to counteract this overactivity by reducing cholesterol efflux. Accordingly, n-3 supplementation and exercise are complementary peroxisomal stimulators and have value in BC prevention [39].

In addition to fully intact phospholipids, we also measured lyso-PtdChos. These levels were decreased in postoperative BC patients. The lyso-PtdCho level has been reported to be decreased in several cancers [40]. We also observed low lyso-PtdCho levels in our study of pancreatic cancer [17]. In addition, high baseline levels of lyso-PtdCho are associated with a reduced risk of future incidence of breast, prostate, and colon cancer [41]. Lyso-PtdCho is a metabolic product generated from PtdCho by phospholipase A2 (PLA2) or lecithin-cholesterol acyltransferase (LCAT) [42]. Alternatively, lyso-PtdCho is consumed by the activity of lyso-PtdCho acyltransferase 1 (LPCAT1). Increased turnover of PtdCho and higher consumption of lyso-PtdCho were observed in malignant tumor cells [43]. The overexpression of LPCAT1 has been described in several cancers [44] and the LPCAT1 protein was also significantly upregulated in primary breast carcinoma tissues compared with normal breast tissues [44,45]. LPCAT1 is the most important enzyme in membrane biogenesis in the lipid remodeling pathway referred to as the Lands cycle and was recently isolated [46]. Our results are consistent with these reports.

The main strength of this study includes the discovery and validation study phase, and the changes in the same patients’ pre-surgery and post-surgery status. Our study also has some limitations. Post-surgery follow-up is only one month later. It is necessary to demonstrate how EtnPls and lyso-PtdCho levels change by long-term post-surgery follow-up. Next, there were a relatively small number of participants and they were only Japanese, which may limit the generalizability of our findings. Therefore, the involvement of large trials from multiple ethnic groups is needed to better assess breast cancer and the metabolic signatures of lipids.

## 4. Materials and Methods

### 4.1. Study Cohorts

Figure 8 provides a summary of the present study, including the discovery phase in study 1 (identify dysregulated lipid metabolic systems with BC among 2000 unique sample-specific accurate masses) and the validation phase (investigate the selected lipid metabolites in both studies, to confirm the reproducibility in study 1, and to validate in another population in study 2). Pre-surgery blood samples of BC patients were collected in Kyoto Prefectural University of Medicine between May 2017 and December 2017, including 69 samples assessed in study 1 and between October 2018 and July 2019, including 62 samples assessed in study 2. We collected 31 samples one month after surgery as post-surgery samples of 62 BC patients in study 2. The study protocol was approved by the Kyoto Prefectural University of Medicine Medical Ethics Board (approval no. ERB-C-1279 all study subjects). Blood samples of healthy controls were collected during the Japan Multi-Institutional Collaborative Cohort (J-MICC) study [47,48]. We randomly selected 48 samples for study 1 and 121 samples for study 2. The subjects were considered cancer-free based on clinical examination and interview. All samples were processed consistently and stored at −80 °C until analysis.

### 4.2. Sample Extraction

Plasma samples were extracted using a modified version of the protocol described by Goodenowe et al. [12]. Briefly, 10 µL of serum was diluted with 50 µL of 0.1% formic acid and subjected to extraction three times with 1.0 mL of acidified ethyl acetate (98:2 ethyl acetate: 0.1% formic acid). Then, 100 µL of a 95% methanol/water solution was added to 400 µL of the combined ethyl acetate fractions and stored at −80 °C prior to analysis.

### 4.3. Mass Spectrometry Analysis

Extracts were directly injected into a Thermo Fisher Scientific LTQ Orbitrap mass spectrometer (Thermo Fisher Scientific, Waltham, MA, USA) in both positive and negative ionization electrospray modes at a flow rate of 200 µL/min. Full scan mass spectral data were collected for masses of 150–1200 amu at maximum resolution. A common pooled reference serum sample was prepared before the study and aliquots of this pooled serum were extracted with each batch of study samples and run on the mass spectrometer at the beginning, middle, and end of each run batch to monitor and correct for batch-to-batch variance throughout the study. Phospholipid species of interest were identified based upon their [M − H]^−^ or [M]^+^ or [M + H]^+^ accurate masses (mass accuracy < 1 ppm). Only accurate mass species represented by single Gaussian peaks subjected to baseline resolution from any surrounding mass peaks were included in the analyses. Typically, a minimum of 5 scans per peak is needed to obtain a reproducible result. In all mass spectrometers operated under standard operating conditions, the signal is proportional to the analyte concentration over 3–5 orders of magnitude, which makes mass spectrometry a popular analytical tool. The intensity of each species was determined by averaging 20 contiguous scans to ensure a robust measurement process.

### 4.4. Data Analyses

The relationship between the absolute abundances of each species and diagnosis and treatment was performed and resulted in the observation that the phospholipid changes were species-specific with some species not changing and the magnitude of changes in the species that were changing was also species-specific (Appendix A). To reduce within-person biological variability, a representative unaffected species (18:1-containing species) was used to generate relative abundance ratios for all of the species studied. This improved the precision (*p*-value) of the observations. Specifically, of each person, we divided all FFA by FFA18:1, all lyso-PtdCho by lyso-PtdCho 18:1, all PtdEtn species by PtdEtn 18:0/18:1, and all PtdCho by PtdCho 16:0/18:1 (Figure 1 and Figure 4). To further illustrate and compare the selective depletion of 18:2 species and EtnPls in BC, representative pairs of PtdEtn and EtnPls species were normalized to PtdEtn 18:0/18:1 (Figure 2 and Figure 3). We describe BC means as relative intensity when control means were 1.00 to articulate the differences.

### 4.5. Statistical Analyses

Continuous variables are expressed as means, and categorical data are expressed as frequencies and proportions. Inter-group comparisons were performed using Welch’s *t*-tests for continuous variables. The diagnostic performance (accuracy) of the models was assessed via receiver operating characteristic (ROC) curve analysis. For comparative purposes, the corresponding areas under the curve (AUC) were calculated and reported. The statistical analyses were performed using R version 3.2.3 (http://www.r-project.org, accessed on 24 July 2021), SPSS software, version 27 (IBM Japan, Tokyo, Japan), and STATA ver.15 (Stata Corp, College Station, TX, USA). All statistical tests were two-tailed, and analysis items with *p* < 0.05 were considered statistically significant.

## 5. Conclusions

Low EtnPls and lyso-PtdCho levels are robust predictors of BC risk that persist postoperatively. The overactive fatty acid elongase (as measured by a selective depletion of 18:2 containing phospholipids) was a predictor of active tumor presence and returned to normal post-surgery. These results indicate that it may be possible to screen for both BC risk and BC activity using a simple blood test.

## Figures and Tables

**Figure 1 cancers-13-04170-f001:**
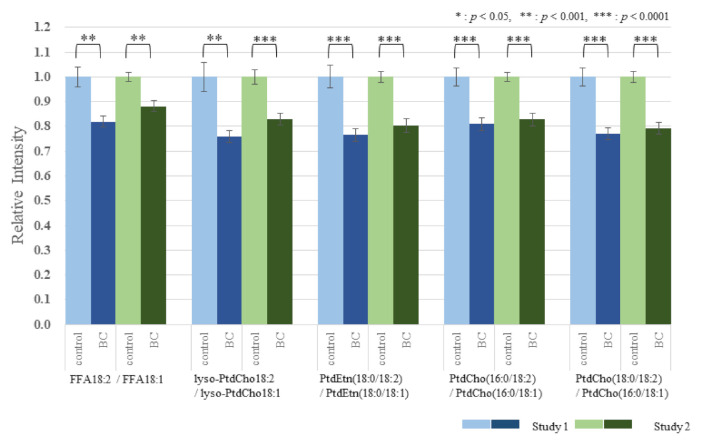
Similar results in both the discovery and validation study, the lower ratio of each species containing 18:2 to each species containing 18:1 (18:2/18:1 ratio) across multiple lipid pools in pre-surgery BC relative to the 18:2/18:1 ratios in control. (BC means are described as relative intensity when control means were 1.00).

**Figure 2 cancers-13-04170-f002:**
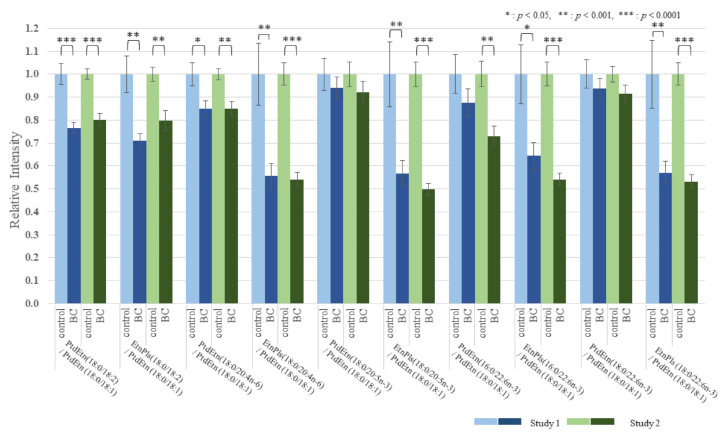
Selected PtdEtn and EtnPls species normalized to PtdEtn 18:0/18:1. (Control means as 1.00 vs. BC means as relative intensity to control).

**Figure 3 cancers-13-04170-f003:**
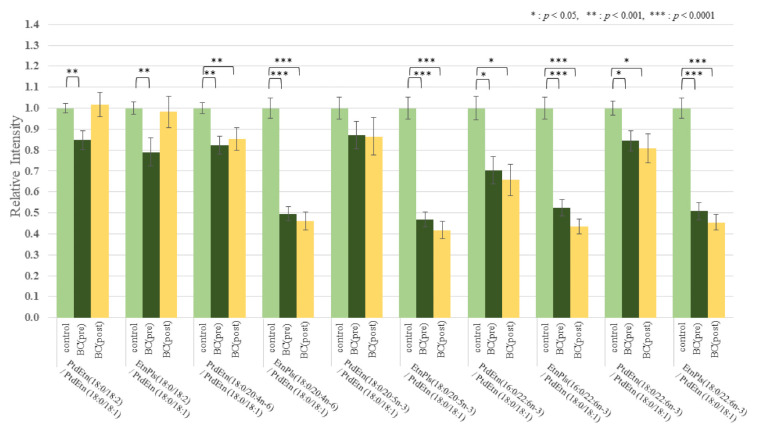
Selected PtdEtn and EtnPls species normalized to PtdEtn 18:0/18:1 in study 2. (Control means as 1.00 vs. pre- or post-surgery BC means as relative intensity to control).

**Figure 4 cancers-13-04170-f004:**
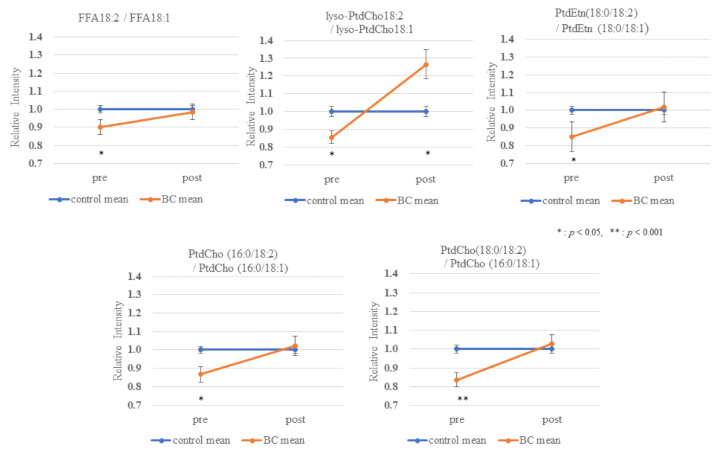
Intensity of the 18:2/18:1 ratios of pre and post-surgery BC relative to the 18:2/18:1 ratios of control in study 2. (Control means as 1.00 and BC means as relative intensity to control).

**Figure 5 cancers-13-04170-f005:**
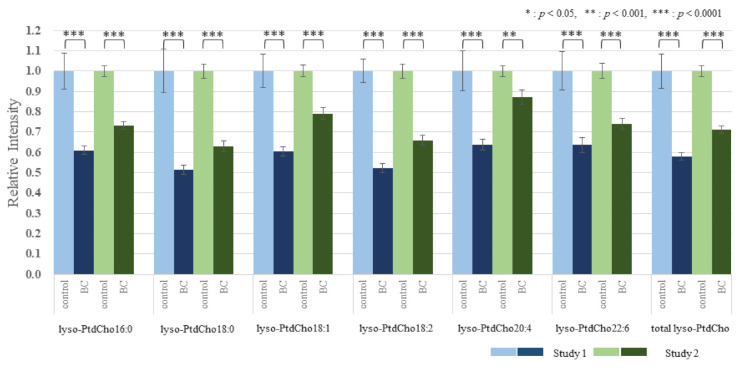
Total species of lyso-PtdCho in pre-surgery BC in both studies. (Control means as 1.00 vs. BC means as relative intensity to control).

**Figure 6 cancers-13-04170-f006:**
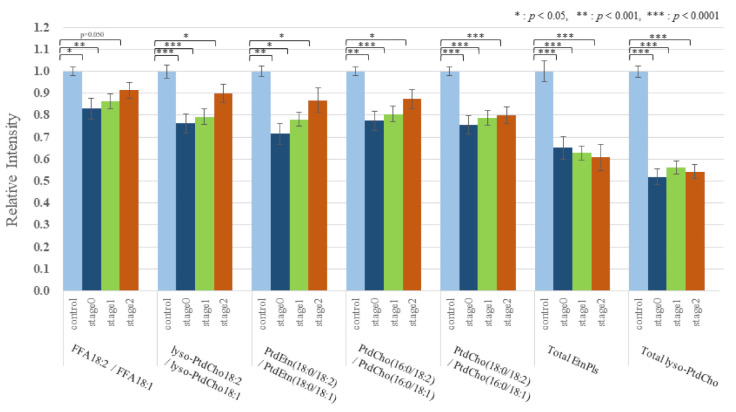
Intensity of the 18:2/18:1 ratios of each stage relative to control in study 2. (Control means as 1.00 and BC means as relative intensity to control).

**Figure 7 cancers-13-04170-f007:**
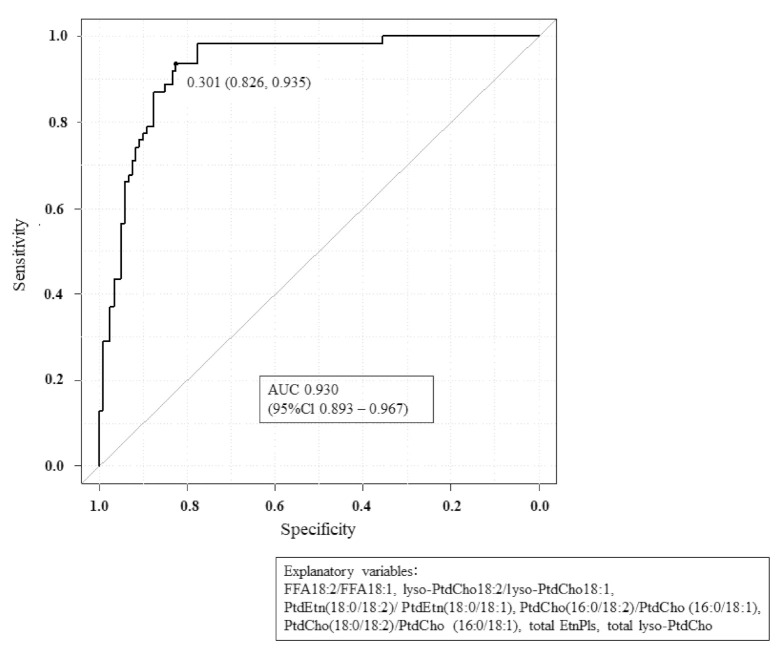
The ROC curve of control vs. BC in study 2 with AUC = 0.930 is described by the combination of 18:2/18:1 series, total EtnPls, and total lyso-PtdCho.

**Figure 8 cancers-13-04170-f008:**
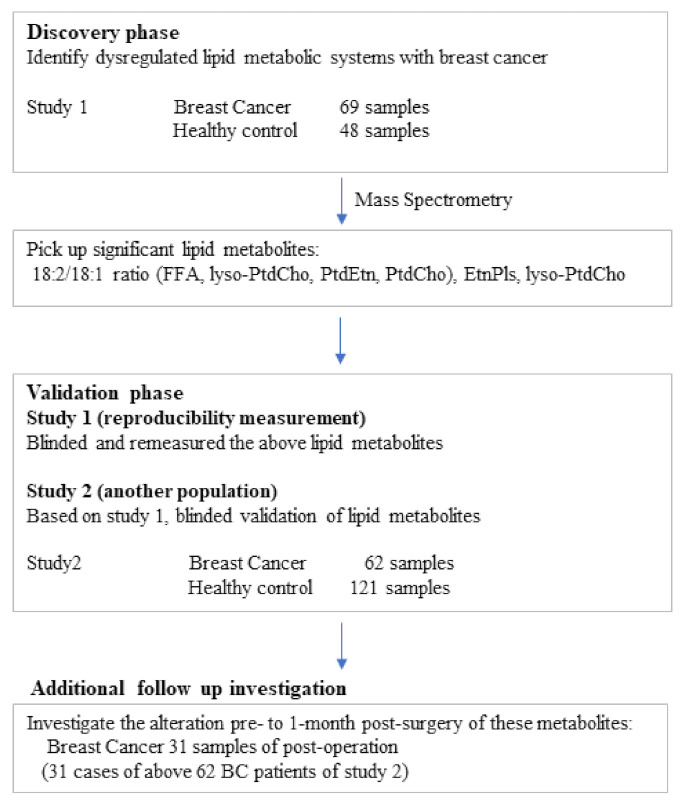
Flow chart of the study participants.

**Table 1 cancers-13-04170-t001:** Characteristics of study 1 and study 2.

	Study 1		Study 2	
	Breast Cancer	Control	Breast Cancer	Control
Total	69	48	62	121
Age (SD)	59.84 (13.70)	61.58 (9.07)	59.59 (13.79)	59.26 (13.32)
(Range)	(34–91)	(41–74)	(34–78)	(39–74)
Stage 0 (*n*)	26.1% (18)	-	22.6% (14)	-
Stage 1 (*n*)	42.0% (29)	-	41.9% (26)	-
Stage 2 (*n*)	20.3% (14)	-	33.9% (21)	-
Stage 3 (*n*)	5.8% (4)	-	1.61% (1)	-
Stage 4 (*n*)	5.8% (4)	-	0% (0)	-

## Data Availability

The data presented in this study are available on request from the corresponding author.

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
