# Peer review of "Plasmalogen Deficiency and Overactive Fatty Acid Elongation Biomarkers in Serum of Breast Cancer Patients Pre- and Post-Surgery—New Insights on Diagnosis, Risk Assessment, and Disease Mechanisms"

_cancers, 2021, doi:10.3390/cancers13164170_

Round 1
Reviewer 1 Report
- Please double-check the wording in the manuscript. For instance, “ A meta-analysis of prospective cohort studies failed to assess the relation between serum fatty acid levels and the risk of BC due to the relatively limited number of studies” -->detect may be a better choice than assess.
- The main finding seemed to be “persons with BC have a lower 158 18:2/18:1 ratio, lower PlsEtn levels, and lower lyso-PtdCho levels than persons without BC”. Can people use “low PlsEtn and lyso-PtdCho levels” for early detection of BC? Or is there any suggestion for the readers in terms of early detection?
- It is suggested to discuss the clinical implication regarding the finding of “the 18:2/18:1 ratios returned to normal following surgical tumor removal.” How can readers use this information in regards to predicting prognosis?
Author Response
Thank you for reviewing our manuscript (cancers-1256330). We appreciate your helpful suggestions and comments. We have addressed your concerns as follows;
- Please double-check the wording in the manuscript. For instance, “ A meta-analysis of prospective cohort studies failed to assess the relation between serum fatty acid levels and the risk of BC due to the relatively limited number of studies” -->detect may be a better choice than assess.
Response:
As suggested, we have revised the sentence as follows (page 4, lines 79): ”A meta-analysis of prospective cohort studies found no association between serum fatty acid levels and risk of breast cancer [7].”
- The main finding seemed to be “persons with BC have a lower 18:2/18:1 ratio, lower PlsEtn levels, and lower lyso-PtdCho levels than persons without BC”. Can people use “low PlsEtn and lyso-PtdCho levels” for early detection of BC? Or is there any suggestion for the readers in terms of early detection?
Response:
This study results indicate that it may be possible to enable early detection of BC using a simple blood test. In general, mammography and ultrasonography are commonly used for BC screening. If blood test can detect BC with high accuracy, it is lower the bar for BC screening. It can also be expected to improve the BC screening rate.
- It is suggested to discuss the clinical implication regarding the finding of “the 18:2/18:1 ratios returned to normal following surgical tumor removal.” How can readers use this information in regards to predicting prognosis?
Response:
In the postoperative follow-up, monitoring of the 18:2/18:1 ratios may enable early detection of recurrence and occurrence of breast cancer. However, this study alone is difficult to prove that and requires further studies.
Your detailed review and comments have been very helpful in enabling us to revise our manuscript. We believe that our manuscript has been improved and strengthened by the changes made based on your comments.
Reviewer 2 Report
The manuscript reports the results from a comprehensive lipid analysis of the serum of subjects to determine if metabolic (in fact lipid metabolites) signatures of over-active fatty acid elongation and other lipid changes could be detected in BC versus non-BC subjects and post-surgery BC subjects. The results have important potential implications for BC screening.
I’ve enjoyed reviewing the manuscript and consider it a valuable contribution to breast cancer metabolic studies. The authors purport to have performed “a comprehensive metabolomic analysis of serum (…)” when in fact only fatty acids and lipids species containing fatty acids were analysed. This must be corrected in the text. The tables should also be revised since their format is somewhat clumsy and don’t convey important information. The authors should mention “lipid metabolites” instead of “metabolites”.
When explaining the discovery and validation studies, the authors mention in the methods (line 265) “discovery phase (study 1) and the validation phase (study 2)”, and in line 88 it is said “All discovery samples along with all study 2 samples, which were sent to the laboratory blinded (…)” while in the caption of Figure 4 it is written “in study 2 (control versus pre- and post-surgery).” There seems to be conflicting information as to what is study 1 and study 2 and discovery and validation: either both studies 1 and 2 are of control vs. BC or study 2 includes also post-surgery data. This should be made very clear; I don’t find the flow chart, as presented, helpful.
Detailed comments (I have made further comments directly on the pdf file):
I don’t agree with the authors notation for fatty acids, the authors should use a notation that clarifies what type of fatty acids are being analysed giving information on the carbon number, number of insaturations and place of first insaturation, a notation like 18:2w6 or 22:6w3 (or e.g. 22:6n-3) would be most helpful. The notation should be used throughout the text to avoid making clumsy reference to fatty acids (FA) “omega-3 docosahexaenoic acid (DHA, 22:6)” (line 105) and “omega-3 DHA (22:6)” (line 171); in line 171, mentioning 22:6w3 would be enough.
There are some problems with the tables:
Table 1 – Spaces should be added between number or % and left parenthesis. In the legend, the authors state “this study” when previously mentioned they were reporting the results from “two case control studies” (abstract). Please revise. There is lack of explanation for row under Age (SD), is it Age interval? I also dislike the use of lower case letters in headings for columns and rows.
Table 2. This table should be presented as supplement information or in the section Materials and Methods; I don’t consider it to be a result from the experiments. The authors should state in the caption that they are presenting phospholipid species, plasmalogens and fatty acids assessed in the analyses. As the lipid metabolites are again listed in Table 3, the authors may simply refer that table instead of table 2. It is not clear why the authors use a full and short-hand notation (also this information is repeated in tables 2 and 3) when the full notation seems to be presented throughout the text and short hand notation in figures.
Table 3. The format of caption and table is not correct. This table is difficult to read and is better suited to be presented as a heat map. The caption does not explain the values (units?) presented in the table; in the caption the authors should state they are presenting abundance data (?) of phospholipid species, plasmalogens and fatty acids.
In the sections 2.2. and 2.3 the authors mention “biochemical system changes” and “serum biochemistry” when in fact they only assessed variations in lipid metabolites, that should be clarified.
The authors show multiple figures with significant values for a statistical test that is not mentioned.
Figure 5 format should be revised, and I don’t understand why in the caption it says that these figures concern C18:2/C18:1 ratios; the authors show the ratio of comparable lipid species containing C18:2 and C18:1 fatty acids.
Materials and Methods
Why is figure 1 presented as Figure 1 after the figures of previous sections?
The authors perform a Welch's t-test, or unequal variances t-test, to compare inter-group significant differences. Multiple comparisons are performed for phospholipid species and these should be adjusted for multiple testing using false discovery rate (FDR q-values)
Author Response
Thank you for reviewing our manuscript (cancers-1256330). We appreciate your helpful suggestions and comments. We have addressed your concerns as follows;
- The authors purport to have performed “a comprehensive metabolomic analysis of serum (…)” when in fact only fatty acids and lipids species containing fatty acids were analysed. This must be corrected in the text. The tables should also be revised since their format is somewhat clumsy and don’t convey important information. The authors should mention “lipid metabolites” instead of “metabolites”.
Response:
As suggested, we have revised “lipid metabolites” instead of “metabolites” in the manuscript, Tables and Figures.
- When explaining the discovery and validation studies, the authors mention in the methods (line 265) “discovery phase (study 1) and the validation phase (study 2)”, and in line 88 it is said “All discovery samples along with all study 2 samples, which were sent to the laboratory blinded (…)” while in the caption of Figure 4 it is written “in study 2 (control versus pre- and post-surgery).” There seems to be conflicting information as to what is study 1 and study 2 and discovery and validation: either both studies 1 and 2 are of control vs. BC or study 2 includes also post-surgery data. This should be made very clear; I don’t find the flow chart, as presented, helpful.
Response:
As suggested, there was a complicated and difficult expression about study design.
We have revised the sentence in the Methods section as follows (page 13, lines 257-261): “Figure 8 provides a summary of the present study, including the discovery phase in study 1 (identify dysregulated lipid metabolic systems with BC among 2000 unique sample-specific accurate masses) and the validation phase (investigate the selected lipid metabolites in both studies, to confirm the reproducibility in study 1, and to validate in another population in study 2). Pre-surgery blood samples of BC patients were collected in Kyoto Prefectural University of Medicine between May 2017 and December 2017, including 69 samples assessed in study 1 and between October 2018 and July 2019, including 62 samples assessed in study 2. We collected 31 samples one month after surgery as post-surgery samples of 62 BC patients in study 2.”
Along with that, the flow chart (figure 8) has been modified.
- Detailed comments (I have made further comments directly on the pdf file):
Response:
We have revised the manuscript using red-colored text according to your comments in the pdf file.
- I don’t agree with the authors notation for fatty acids, the authors should use a notation that clarifies what type of fatty acids are being analysed giving information on the carbon number, number of insaturations and place of first insaturation, a notation like 18:2w6 or 22:6w3 (or e.g. 22:6n-3) would be most helpful. The notation should be used throughout the text to avoid making clumsy reference to fatty acids (FA) “omega-3 docosahexaenoic acid (DHA, 22:6)” (line 105) and “omega-3 DHA (22:6)” (line 171); in line 171, mentioning 22:6w3 would be enough.
Response:
The following changes have been made to unify the notation; omega-3 (n-3), omega-6 (n-6), oleic acid (18:1), linoleic acid (18:2), docosahexaenoic acid (22:6n-3), arachidonic acid (20:4n-6).
- Table 1 – Spaces should be added between number or % and left parenthesis. In the legend, the authors state “this study” when previously mentioned they were reporting the results from “two case control studies” (abstract). Please revise. There is lack of explanation for row under Age (SD), is it Age interval? I also dislike the use of lower case letters in headings for columns and rows.
Response:
As suggested, we have revised in Table 1.
- Table 2. This table should be presented as supplement information or in the section Materials and Methods; I don’t consider it to be a result from the experiments. The authors should state in the caption that they are presenting phospholipid species, plasmalogens and fatty acids assessed in the analyses. As the lipid metabolites are again listed in Table 3, the authors may simply refer that table instead of table 2. It is not clear why the authors use a full and short-hand notation (also this information is repeated in tables 2 and 3) when the full notation seems to be presented throughout the text and short hand notation in figures.
Response:
As suggested, Tables 2 and 3 had confusing expressions.
If we write full annotation in the figures, it will be too much words and be difficult to understand, so we thought that shorthand is necessary to maintain the visibility of the figures.
We have revised in Table 2 and Table 3 (changed to Supplemental table1) to make them easier to read.
- Table 3. The format of caption and table is not correct. This table is difficult to read and is better suited to be presented as a heat map. The caption does not explain the values (units?) presented in the table; in the caption the authors should state they are presenting abundance data (?) of phospholipid species, plasmalogens and fatty acids.
Response:
“Mean Intensity Normalized to Respective Control” indicates the ratio, so there is no unit.
“TTEST to Respective Control (p)” indicates the p-value using welch's t-tests.
In this manuscript, figures are used efficiently to make it easier for the reader to understand the results. Table 3 shows the analysis values. In sum, Table 3 is supplementary information for figure 1-6. We have modified Table 3 to make it easier to understand and changed to Supplemental table 1 to avoid reader’s confusion.
- In the sections 2.2. and 2.3 the authors mention “biochemical system changes” and “serum biochemistry” when in fact they only assessed variations in lipid metabolites, that should be clarified.
Response:
As suggested, we have revised “biochemical system changes” to “lipid metabolomic system changes”, and “serum biochemistry” to “lipid metabolomic system”.
- The authors show multiple figures with significant values for a statistical test that is not mentioned.
Response:
We consistently analysed control and between each group with t-test.
We showed that there are significant differences in the figures, and the measured values were shown in Supplemental table 1.
- Figure 5 format should be revised, and I don’t understand why in the caption it says that these figures concern C18:2/C18:1 ratios; the authors show the ratio of comparable lipid species containing C18:2 and C18:1 fatty acids.
Response:
We have changed Figure 5 legend to “The each species containing18:2/ each species containing 18:1 ratios of pre and post-surgery of study 2.” We also have revised “each species containing18:2/ each species containing 18:1” instead of “C18:2/C18:1” in the manuscript and Figure legend.
- Why is figure 1 presented as Figure 1 after the figures of previous sections?
Response:
We have revised “Figure 1” to “Figure 8”.
- The authors perform a Welch's t-test, or unequal variances t-test, to compare inter-group significant differences. Multiple comparisons are performed for phospholipid species and these should be adjusted for multiple testing using false discovery rate (FDR q-values)
Response:
As mentioned in comments 9, we consistently analyze control and between each group with t-test. Multi-group comparisons like ANOVA are not performed.
Your detailed review and comments have been very helpful in enabling us to revise our manuscript. We believe that our manuscript has been improved and strengthened by the changes made based on your comments.
Reviewer 3 Report
In the manuscript entitled “Plasmalogen deficiency and overactive fatty acid elongation biomarkers in serum of breast cancer patients pre- and post-surgery – new insights on diagnosis, risk assessment and disease mechanisms”, the authors investigated metabolomic fatty acid analysis of serum from two case control studies of non-BC, BC subjects pre-surgery and BC subjects one month post-surgery to determine if metabolic signatures of over-active fatty acid elongation and other lipid changes could be detected in BC (pre and post-surgery) versus non-BC subjects. Although, the manuscript language is clear, but it is hard to follow. Below are some comments that may hopefully help improve the manuscript.
-Line 52, “A meta-analysis of prospective cohort studies failed to assess the relation between serum fatty acid levels and the risk of BC due to the relatively limited number of studies 53 [7]” actually the meta-analysis did not fail to “assess” the relation between serum fatty acids and BC risk. This meta-analysis (of prospective studies from 1980’s till 2015) results actually suggested that dietary total fat and fatty acids might not be associated with the risk of breast cancer. Please rephrase your sentence and discuss your results considering this meta-analysis.
-Lines 58-59, “Yamashita et al. reported that the level of the n-6 family C18:2 was lower in BC tissues and believed that ELOVL5 accelerated elongation [11]”. Please specify the “n-6 family C18:2” by naming specific fatty acids (for example AA). Also, if ELOVL5 activity is enhanced, don’t you expect elongation products be increased? Please comment.
-Line 60, “The researchers also reported that the mRNA expressions of ELOVL1, 5, and 6 were significantly higher in BC tissues than in corresponding normal breast tissues” please add references.
-Line 62, “We have previously described low levels of plasmalogens and ultra-long chain polyunsaturated fatty acids” please name the fatty acids you assessed in your studies.
-Lines 74-78, “Accurate masses corresponding to specific species of phosphatidylethanolamine (PtdEtn), phosphatidylcholine (PtdCho), ethanolamine plasmalogens (PlsEtn), lyso-PtdCho, and free fatty acids (FFA) were observed to have significantly different levels in BC versus non-BC persons (data not shown)” why is this data not provided?
-Table 1, discovery is misspelled.
-Table 2, what do you mean by M-H column (is this protonated ion mass?) but what does it mean when you state fatty acid ratios as for example in 18:0/18:1 and does the fatty acid mass change depending on the phospholipid fraction? for example, 18:1 positive ion mass under phosphatidylcholine is 522.3554 and it is 281.2486 under free fatty acids.
-Table 2 is a bit crowded. Please remove the shorthand column and stick to the full abbreviated annotation. Also, you do not need to repeat the phospholipid name before each fatty acid under each phospholipid fraction, please remove it.
-Table 3 is a bit crowded. Please remove the shorthand column. Further, it is stated that data “Normalized to Respective Control” do you mean that the fatty acid ratios in BC were divided by the ratios in control e.g. BC (18:0/18:1)/control (18:0/18:1)? And why not use a more direct ratio as for example 18:2 in BC/18:1 in control? Why all fatty acids in Phosphatidylethanolamines (PtdEtn) fraction expressed as ratios to C18:00 except /22:6 was to 16:0. The same goes for figure 4.
-Table 3, Why were the data of control samples (from study 1 and 2) compiled while BC data were not compiled? Do the comparisons of each study (with separate control) give similar results without compiling controls? What the control presented in figure 2 represent (compiled samples from study 1 and 2).
-There is no need for figure 5 (the point is already shown in table 3 and figure 4).
-Line 268, “2017/5-and December 2017, including 69 …………” change 5 to May.
-Line 270 “We collected 31 samples one month after surgery as postoperative samples of 62 BC patients” how come 31 samples were collected from 62 BC patients!!
-Provide the full name of the abbreviation “FI-FTMS”.
-Figure 2, why the last 3 sets of comparisons are expressed so differently for example in pe fraction the ratio is expressed as 18:0/18:2/18:0/18:1 instead of a direct 18:2/18:1 ratio (as in fa and lpc fractions). The same goes for pc fraction. Is the measurement in pc presented twice?!
Author Response
Thank you for reviewing our manuscript (cancers-1256330). We appreciate your helpful suggestions and comments. We have addressed your concerns as follows;
- -Line 52, “A meta-analysis of prospective cohort studies failed to assess the relation between serum fatty acid levels and the risk of BC due to the relatively limited number of studies 53 [7]” actually the meta-analysis did not fail to “assess” the relation between serum fatty acids and BC risk. This meta-analysis (of prospective studies from 1980’s till 2015) results actually suggested that dietary total fat and fatty acids might not be associated with the risk of breast cancer. Please rephrase your sentence and discuss your results considering this meta-analysis.
Response:
As suggested, we have revised the sentence as follows (page 4, lines 79): ”A meta-analysis of prospective cohort studies found no association between serum fatty acid levels and risk of breast cancer [7].”
- -Lines 58-59, “Yamashita et al. reported that the level of the n-6 family C18:2 was lower in BC tissues and believed that ELOVL5 accelerated elongation [11]”. Please specify the “n-6 family C18:2” by naming specific fatty acids (for example AA). Also, if ELOVL5 activity is enhanced, don’t you expect elongation products be increased? Please comment.
Response:
The following changes have been made to unify the notation; omega-3 (n-3), omega-6 (n-6), oleic acid (18:1), linoleic acid (18:2), docosahexaenoic acid (22:6n-3), arachidonic acid (20:4n-6).
If ELOVL5 activity is enhanced, elongation products may be increased. Actually, Yamashita et al. showed an increase in elongated products such as C16. However, this phenomenon is found in breast cancer tissue. Our study used serum sample. The decrease in 22: 6n-3 and 20: 4n-6 is due to peroxisome dysfunction, which is a risk factor for breast cancer, and is considered to be a different phenomenon from enhanced ELOVL5 activity in breast cancer tissue. In this study, at one month post-surgery, the 18:2/18:1 ratios increased versus pre-surgery and were no longer different from non-BC subjects. In contrast to the elongation biomarkers, docosahexaenoic acid (22:6n-3) containing ethanolamine plasmalogen (PlsEtn) species were observed to be further decreased in BC subjects one month post-surgery versus pre-surgery levels.
- -Line 60, “The researchers also reported that the mRNA expressions of ELOVL1, 5, and 6 were significantly higher in BC tissues than in corresponding normal breast tissues” please add references.
Response:
We have added reference [11] at the sentence.
- -Line 62, “We have previously described low levels of plasmalogens and ultra-long chain polyunsaturated fatty acids” please name the fatty acids you assessed in your studies.
Response:
As suggested, we have revised the sentence as follows (page 5, lines 93-97): ”We have previously described the low level of 28 carbon polyhydroxylated and polyunsaturated long-chain fatty acid (C28H46O4), named GTA-446 in coloractal cancer patients and the low level of 36 carbon polyhydroxylated and polyunsaturated long-chain fatty acid (C36H66O6), named PTA-594 in pancreatic cancer patients [12-15]. We also described low levels of plasmalogens in both patients [12-15].”
- -Lines 74-78, “Accurate masses corresponding to specific species of phosphatidylethanolamine (PtdEtn), phosphatidylcholine (PtdCho), ethanolamine plasmalogens (PlsEtn), lyso-PtdCho, and free fatty acids (FFA) were observed to have significantly different levels in BC versus non-BC persons (data not shown)” why is this data not provided?
Response:
The discovery phase (study 1) serum samples resulted in over 2000 unique sample-specific accurate masses. Since it is not reader-friendly to publish all the results, we have picked up significant lipid metabolites: C18:2/C18:1(FFA, Lyso-PtdCho, PtdEtn, PtdCho), PlsEtns, Lyso-PtdChos.
- -Table 1, discovery is misspelled.
Response:
We have removed “discovery” and “validation” in Table 1.
- -Table 2, what do you mean by M-H column (is this protonated ion mass?) but what does it mean when you state fatty acid ratios as for example in 18:0/18:1 and does the fatty acid mass change depending on the phospholipid fraction? for example, 18:1 positive ion mass under phosphatidylcholine is 522.3554 and it is 281.2486 under free fatty acids.
Response:
M-H means the de-protonated version of the neutral mass: M (neutral) minus a proton [H+] = [M-H]-. This is a negatively charged ion. The 18:0/18:1 means the combination of two fatty acids at the sn-1 and sn-2 positions of the glycerol backbone. Technically, the total carbons and double bonds are 36:1. However, since the sn-1 position is highly conserved (>95% is 16:0, 18:0, or 18:1) and saturated fatty acids at sn-2 are rare, a total carbon/double bond number of 36:1 is almost exclusively 18:0 at sn-1 and 18:1 at sn-2. Under free fatty acids, the fatty acid is not attached to a phospholipid so it is just the M-H mass of the fatty acid. The 522 mass above is the mass for a lyso-PC with 18:1 at either sn-1 or sn-2 and a free hydroxy group at the other sn-1 or -2 position.
We have added captions for [M-H]-and [M]+ in Table 2.
- -Table 2 is a bit crowded. Please remove the shorthand column and stick to the full abbreviated annotation. Also, you do not need to repeat the phospholipid name before each fatty acid under each phospholipid fraction, please remove it.
Response:
As suggested, Tables 2 had confusing expressions. The shorthands sre nesessary to maintain the visibility of the figures. If we write full annotation in the figures, it will be too many words and be difficult to understand. To match each shorthands to each full annotations, they cannot be omitted. We have revised in Table 2 and Table 3 (we have changed to Supplemental table1) to make them easier to read.
- -Table 3 is a bit crowded. Please remove the shorthand column. Further, it is stated that data “Normalized to Respective Control” do you mean that the fatty acid ratios in BC were divided by the ratios in control e.g. BC (18:0/18:1)/control (18:0/18:1)? And why not use a more direct ratio as for example 18:2 in BC/18:1 in control? Why all fatty acids in Phosphatidylethanolamines (PtdEtn) fraction expressed as ratios to C18:00 except /22:6 was to 16:0. The same goes for figure 4.
Response:
As mentioned above, we have revised in Table 2 and Table 3 (changed to Supplemental table1).
What the reviewer is interpreting as ratios is NOT ratios, each row in Table 3 (changed to Supplemental table1) represents a specific phospholipid species. The title even states this “The data of individual species…”
We have split CTL-1/2 into CTL-1 and CTL-2, added captions, and modified it to be reader-friendly.
- -Table 3, Why were the data of control samples (from study 1 and 2) compiled while BC data were not compiled? Do the comparisons of each study (with separate control) give similar results without compiling controls? What the control presented in figure 2 represent (compiled samples from study 1 and 2).
Response:
As mentioned above, we have revised in Table 2 and Table 3 (changed to Supplemental table1).
- -There is no need for figure 5 (the point is already shown in table 3 and figure 4).
Response:
Figure 3 and 4 do not mention FFA, Lyso-PtdCho and PtdCho. Therefore, it cannot be omitted.
- -Line 268, “2017/5-and December 2017, including 69 …………” change 5 to May.
Response:
As suggested, we have revised.
- -Line 270 “We collected 31 samples one month after surgery as postoperative samples of 62 BC patients” how come 31 samples were collected from 62 BC patients!!
Response:
Only 31 samples were collected as follow-ups one month postoperative.
- -Provide the full name of the abbreviation “FI-FTMS”.
Response:
FI-FTMS is “Flow Injection - Fourier Transform Mass Spectrometry”.
We changed “FI-FTMS” to “Mass Spectrometry” to make it the same as the manuscript.
- -Figure 2, why the last 3 sets of comparisons are expressed so differently for example in pe fraction the ratio is expressed as 18:0/18:2/18:0/18:1 instead of a direct 18:2/18:1 ratio (as in fa and lpc fractions). The same goes for pc fraction. Is the measurement in pc presented twice?!
Response:
For example, the central pe(18:0/18:2)/pe(18:0/18:1) indicates that PtdEtn18:0/18:2 is divided by PtdEtn 18: 0/18: 1. The same applies to the other two.
Your detailed review and comments have been very helpful in enabling us to revise our manuscript. We believe that our manuscript has been improved and strengthened by the changes made based on your comments.
Round 2
Reviewer 2 Report
Comments and recommendations:
The authors presented a revised the manuscript according to recommendations, although I consider that some issues have not been addressed suitably.
One important point is what is being measured; I assumed that the authors measured abundance levels of lipid metabolites in the serum. In line 282, the authors state that “The intensity of each species was determined by averaging 20 contiguous scans.” but it isn’t stated anywhere that intensity of species measured reflects the abundance of the species in the serum, this should be clearly stated since this is of paramount importance to later propose this method as an alternative to other diagnostic procedures in BC screenings. This study has a high scientific value and the authors should thrive to present their results in a very clear way and avoid ambiguity.
There is still much imprecision regarding how the lipid species are named, and this must be corrected. For instance, PLE (phosphatidylethanolamine?) is written in line 12 without saying what it is, and then the authors name the species as “PtdEtn” in the next line and “pe” in the figures 2 and 3 and again “PtdEtn” in figure´s captions. All names must be revised and multiples acronyms for the same lipid species should be avoided.
In the methods, under statistical analysis, it is not clear if any normalization was applied prior to data statistical analyses and in fact only “means” are mentioned (line 285). In line 112, you do mention that “representative pairs of PtdEtn and PlsEtn species were normalized to PtdEtn 18:0/18:1”. I assume this normalization is done using PtdEtn 18:0/18:1 in the control but that is not stated, and it must be stated. In the caption of figure 3, it is stated that “Selected PtdEtn and PlsEtn species normalized to PtdEtn 18:0/18:1 in study 2 (control versus pre- or post-304 surgery).”, frankly the authors lost me here, is it the same procedure as stated in line 112? If so, why is it that in the XX axis (ref) of BC is above 1? Please be very clear regarding all numerical procedures and state what is done. In the YY axis of figures, you state “relative intensity” but we don’t understand do what is relative (is it to control?, to what lipid species?). From the graphs, I “infer” that all BC intensities are relative to control for the corresponding lipid species, this should be clear and should be mentioned in the “methods” as the general procedure applied to all lipid markers. In the supplementary tables, it is stated “Mean Intensity Normalized to Respective Control”. For instance, in lines 114- 115 it says “The level of PtdEtn 18:0/18:1 was unchanged in BC and non-BC subjects; however, both the levels of PtdEtn 18:0/18:2 and PlsEtn 18:0/18:2 species were decreased in BC subjects”; it should be added “(…)were decreased in BC subjects relative to control”.
As another example, the caption of Figure 1 is incomprehensible, also what does “18:2/” mean?
Sentences like (line 173) “The PlsEtn levels of 20:4n-6 and 22:6n-3 were observed to be consistently lower than the PtdEtn levels of these fatty acids in BC (Figure 3) should be avoided, because it is unclear what it means. Is it “the levels of PlsEtn containing 20:4n-6 and 22:6n-3 FA are consistently lower in BC subjects relative to control”?
I consider it appropriate to present Table 1 in the main text, yet I consider that Table 2 is more suited as supplementary data.
Figure 8 is clearer now, yet the authors should avoid using casual expressions: When you say “pick up” is it “selection”? You should also supply here the “selection” criterion. In the text, you should state the criterion used for this selection (was it a % variation in BC samples relative to control, statistical analysis?). From table 2, we assume some test was made since it is written “significantly different levels in BC versus non-BC”. Please state the statistical analysis used (even if data is not shown) and state in the caption or inside the figure that this is done in the discovery phase study. This is of high relevance to ascribe weight to your claims for the validation phase. In the validation process (I think because it is not stated – and it should) you apply Welch’s t-tests for continuous variables. The authors perform a Welch's t-test, or unequal variances t-test, to compare inter-group significant differences. Multiple comparisons are performed for phospholipid species and fatty acids, and these should be adjusted for multiple testing using false discovery rate (FDR q-values). The fact that the authors do not use ANOVA (as stated in their response) does not justify not testing for false positives resulting from their Welch's t-tests. The authors do perform pairwise testing via Welch's t-tests for every pair of "treatments" in their dataset. The authors did obtain several p-values, which must be "adjusted" by the FDR method (or other more conservative if preferred). In this study, the more stringent threshold for p-value would ensure that your findings are indeed statistically significant. You did start your study by restricting comparisons between healthy and BC samples in the discovery phase by determining the most important features to be compared during validation; during validation you should show that your analysis is robust and not the result of false positives resulting for multiple pair comparisons.
When we set a p-value threshold we accept that there is a 5% chance that the result is a false positive (significant difference between control and treatment). This is acceptable for one test alone, yet when we test (by ANOVA or t-test) a large number of metabolites (sometimes 1000 compounds in metabolite studies), this 5% may result in a large number of false positives (e.g. 50 metabolites in 1000). And this is the largely known multiple test problem. Some techniques, such as the Bonferroni correction are too conservative, while the FDR approach is optimised by using characteristics of the p-value distribution to produce a list of q-values. I would recommend its application to the data set.
Details:
I don’t know if the Abstract should be longer or shorter than the “Simple Summary”. Here it is shorter and it seems odd.
You should read the manuscript carefully to detect small oddities. For instance, when you manetion in line 20 “ever-active fatty acid elongation”, do you mean “metabolic signatures resulting from up-regulated fatty acid elongation”?
The reference list should be revised according to “Directions for authors”; the references for websites don’t seem to be according to the directions.
Author Response
For Reviewer 2
Thank you for reviewing our manuscript (cancers-1256330). We appreciate your helpful suggestions and comments. We have addressed your concerns as follows;
- One important point is what is being measured; I assumed that the authors measured abundance levels of lipid metabolites in the serum. In line 282, the authors state that “The intensity of each species was determined by averaging 20 contiguous scans.” but it isn’t stated anywhere that intensity of species measured reflects the abundance of the species in the serum, this should be clearly stated since this is of paramount importance to later propose this method as an alternative to other diagnostic procedures in BC screenings. This study has a high scientific value and the authors should thrive to present their results in a very clear way and avoid ambiguity.
Response:
Sorry for the confusion. During the analysis of metabolite species by liquid chromatography or flow injection mass spectrometry several factors are involved in ensuring that a reproducible measurement of each analyte is obtained. Ultimately the analyte must enter the mass spectrometer (either a chromatographic retention time or a flow injection time window). It takes a specified amount of time for the mass spectrometer to ionize, transfer, and ultimately for the electron multiplier to register a signal. Depending on the mass spectrometer settings this could be very fast (millisecond) or very slow (seconds). Therefore, the time width of the window divided by the scan time determines how many measurements can be obtained from each sample injection. Typically, a minimum of 5 scans per peak is needed to get a reproducible result. In all mass spectrometers operated under standard operating conditions, the signal is proportional to the analyte concentration over 3-5 orders of magnitude, which makes mass spectrometry a popular analytical tool. The sentence referencing 20 contiguous scans was meant to illustrate the robustness of the measurement process.
To clarify these, we have revised the sentence in Materials and Methods, Mass spectrometry analysis as follows (page 15, lines 294)
“Typically, a minimum of 5 scans per peak is needed to get a reproducible result. In all mass spectrometers operated under standard operating conditions, the signal is proportional to the analyte concentration over 3-5 orders of magnitude, which makes mass spectrometry a popular analytical tool. The intensity of each species was determined by averaging 20 contiguous scans to be more robust the measurement process. “
- There is still much imprecision regarding how the lipid species are named, and this must be corrected.For instance, PLE (phosphatidylethanolamine?) is written in line 112 without saying what it is, and then the authors name the species as “PtdEtn” in the next line and “pe” in the figures 2 and 3 and again “PtdEtn” in figure´s captions. All names must be revised and multiples acronyms for the same lipid species should be avoided.
Response:
As suggested, PLE is phosphatidylethanolamine. The abbreviation for ethanolamine plasmalogen has been changed from PlsEtn to EtnPls to make it easier to understand. For Figures, we used shorthand such as pe and pc, but we have matched these with the notation in the text. Therefore, shorthand of Table 2 is no longer needed, so we have deleted shorthand and modified Table2 to Supplement table1.
- In the methods, under statistical analysis, it is not clear if any normalization was applied prior to data statistical analyses and in fact only “means” are mentioned (line 285). In line 112, you do mention that “representative pairs of PtdEtn and PlsEtn species were normalized to PtdEtn 18:0/18:1”. I assume this normalization is done using PtdEtn 18:0/18:1 in the control but that is not stated, and it must be stated.
Response:
In this study, no mathematical transformations or normalizations were performed prior to statistical analyses. Two types of data were interpreted – absolute species abundances and the relative ratios of specific species abundances. All persons were treated identically. PtdEtn 18:0/18:1 is a species that was observed to be the least affected by BC status. “normalization” to PtdEtn 18:0/18:1 was done within each subject to reduce biological variability with each class of species and across different classes of species and to get a more accurate representation of the changes observed. Group normalizations to control means were done to make the visualization of the data easier to interpret. In detail, the control is normalized by the control's PtdEtn 18:0/18:1, and the BC is normalized by BC's PtdEtn 18:0/18:1. To clarify these, we have revised the sentence in Materials and Methods as follows (page 15, lines 299)
“Data analyses
The relationship between the absolute abundances of each species and diagnosis and treatment was performed and resulted in the observation that the phospholipid changes were species-specific with some species not changing and the magnitude of changes in the species that were changing was also species specific. To reduce within-person biological variability, a representative unaffected species (18:1-containing species) was used to generate relative abundance ratios for all of the species studied. This improved the precision (p-value) of the observations. Specifically, of each person, we divided all FFA by FFA18:1, all lyso-PtdCho by lyso-PtdCho 18:1, all PtdEtn species by PtdEtn 18:0/18:1 and all PtdCho by PtdCho 16:0/18:1.
(figure1,4) To further illustrate and compare the selective depletion of 18:2 species and EtnPls in BC, representative pairs of PtdEtn and EtnPls species were normalized to PtdEtn 18:0/18:1. (Figures 2 and 3) We describe BC means as relative intensity when control means were 1.00 to articulate the differences.”
- In the caption of figure 3, it is stated that “Selected PtdEtn and PlsEtn species normalized to PtdEtn 18:0/18:1 in study 2 (control versus pre- or post-304 surgery).”, frankly the authors lost me here, is it the same procedure as stated in line 112?
Response:
Yes, it is. Figure 3, the same procedure as for Figure 2 is performed for each variable of control, BC (pre-), and BC (post-) in study2.
- If so, why is it that in the XX axis (ref) of BC is above 1? Please be very clear regarding all numerical procedures and state what is done.
Response:
The ref of Figure 2 and 3 also showed each value of BC when the control of study 1 and study 2 is 1.0-fold. The notation for “pe(18:0/18:1)/ pe(18:0/18:1)(ref)” at the left end of Figure 2 and 3 is incorrect, and “pe(18:0/18:1)(ref)” is correct. The calculation method is different for pe(18:0/18:1)(ref) and other values. We deleted pe(18:0/18:1)(ref) of Figure 2 and 3 because it would be confusing if left in the figures.
In addition, we have modified PtdEtn18:0/18:1 in supplemental table2 because the values were incorrect.
- In the YY axis of figures, you state “relative intensity” but we don’t understand do what is relative (is it to control?, to what lipid species?).
Response:
We considered the “relative intensity” was the relative strength with the control as 1.0-fold.
- From the graphs, I “infer” that all BC intensities are relative to control for the corresponding lipid species, this should be clear and should be mentioned in the “methods” as the general procedure applied to all lipid markers. In the supplementary tables, it is stated “Mean Intensity Normalized to Respective Control”.
Response:
We have revised Materials and Methods section.
- For instance, in lines 114- 115 it says “The level of PtdEtn 18:0/18:1 was unchanged in BC and non-BC subjects; however, both the levels of PtdEtn 18:0/18:2 and PlsEtn 18:0/18:2 species were decreased in BC subjects”; it should be added “(…)were decreased in BC subjects relative to control”.
Response:
We have added “relative to control”. (page 7, lines 131)
- As another example, the caption of Figure 1 is incomprehensible, also what does “18:2/” mean?
Response:
“18:2/” means each species containing oleic acid (18:1) divided by each species containing 18:2. We have modified to “ratio of each species containing 18: 2 to each species containing oleic acid (18: 1) (18: 2/18: 1 ratio)”, and we have added about these to Materials and Methods section.
- Sentences like (line 173) “The PlsEtn levels of 20:4n-6 and 22:6n-3 were observed to be consistently lower than the PtdEtn levels of these fatty acids in BC (Figure 3) should be avoided, because it is unclear what it means. Is it “the levels of PlsEtn containing 20:4n-6 and 22:6n-3 FA are consistently lower in BC subjects relative to control”?
Response:
We have modified as follows (page 10, lines 186) “the levels of PlsEtn containing 20:4n-6 and 22:6n-3 in BC subjects are consistently lower relative to the levels of PtdEtn containing 20:4n-6 and 22:6n-3 in BC subjects.” PlsEtn is not compared to control, but to PtdEtn. For example, the two graphs on the right of Figure 2 showed the BC of ple(18:0/22:6)/ple(18:0/18:1) is lower than pe(18:0/22:6)/pe(18:0/18:1). Figure 3 has the same content.
- I consider it appropriate to present Table 1 in the main text, yet I consider that Table 2 is more suited as supplementary data.
Response:
We have modified Table 2 to Supplemental table 1, and Supplemental table 1 to Supplemental table 2. We have rearranged the order of lipid species in Supplemental table 2 to match that order in Figures.
- Figure 8 is clearer now, yet the authors should avoid using casual expressions: When you say “pick up” is it “selection”? You should also supply here the “selection” criterion. In the text, you should state the criterion used for this selection (was it a % variation in BC samples relative to control, statistical analysis?). From table 2, we assume some test was made since it is written “significantly different levels in BC versus non-BC”. Please state the statistical analysis used (even if data is not shown) and state in the caption or inside the figure that this is done in the discovery phase study. This is of high relevance to ascribe weight to your claims for the validation phase.
In the validation process (I think because it is not stated – and it should) you apply Welch’s t-tests for continuous variables. The authors perform a Welch's t-test, or unequal variances t-test, to compare inter-group significant differences. Multiple comparisons are performed for phospholipid species and fatty acids, and these should be adjusted for multiple testing using false discovery rate (FDR q-values). The fact that the authors do not use ANOVA (as stated in their response) does not justify not testing for false positives resulting from their Welch's t-tests. The authors do perform pairwise testing via Welch's t-tests for every pair of "treatments" in their dataset. The authors did obtain several p-values, which must be "adjusted" by the FDR method (or other more conservative if preferred). In this study, the more stringent threshold for p-value would ensure that your findings are indeed statistically significant. You did start your study by restricting comparisons between healthy and BC samples in the discovery phase by determining the most important features to be compared during validation; during validation you should show that your analysis is robust and not the result of false positives resulting for multiple pair comparisons.
When we set a p-value threshold we accept that there is a 5% chance that the result is a false positive (significant difference between control and treatment). This is acceptable for one test alone, yet when we test (by ANOVA or t-test) a large number of metabolites (sometimes 1000 compounds in metabolite studies), this 5% may result in a large number of false positives (e.g. 50 metabolites in 1000). And this is the largely known multiple test problem. Some techniques, such as the Bonferroni correction are too conservative, while the FDR approach is optimised by using characteristics of the p-value distribution to produce a list of q-values. I would recommend its application to the data set.
Response:
To clarify the selection criterion in discovery phase, we have added Supplemental Figure1 and have added the sentence in Materials and Methods as follows. (page 6, lines 107)
” Supplemental Figure 1 shows 18:2 (linoeic acid) /18:1 (oleic acid) ratio are lower significant with the p-value < 0.0005 – 0.000001 between BC patients and healthy controls on the red plots; metabolites of FFA, Lyso-PtdCho, PtdEtn and PtdCho series.”
Supplemental Figure 1 indicates that the p-value is sufficient even with Bonferroni correction. Furthermore, we can eliminate the multiple test issue by utilizing two independent cohorts of samples. When similar independent cohorts have similar trends, then the problem of random statistical significances is largely eliminated. In fact, one could argue that the p-values should be multiplied (p=0.02 in study 1 and p=0.01 in study 2 = 0.02 x 0.01 = 0.0002 that 2 independent studies would give the same result. False discovery rates are designed to prevent over fitting of data within a single cohort not across multiple cohorts. Rolling dice and getting 12 (6+6) has a certain randomness to it. Rolling 12 twice in a row is much more unlikely.
- I don’t know if the Abstract should be longer or shorter than the “Simple Summary”. Here it is shorter and it seems odd.
You should read the manuscript carefully to detect small oddities. For instance, when you mention in line 20 “ever-active fatty acid elongation”, do you mean “metabolic signatures resulting from up-regulated fatty acid elongation”?
Response:
Abstract (282 words) is longer than Simple Summary. As suggested, “over-active fatty acid elongation” is means “metabolic signatures resulting from up-regulated fatty acid elongation”.
- The reference list should be revised according to “Directions for authors”; the references for websites don’t seem to be according to the directions.
Response:
We have revised the reference No. 1 and 2.
Your detailed review and comments have been very helpful in enabling us to revise our manuscript. We believe that our manuscript has been improved and strengthened by the changes made based on your comments.
Author Response
Reviewer3's "Comments and Suggestions for Authors
" was blank.
We asked about this Assistant Editor Miss Ana Fenesan on July 15th and got the answer that we only need to respond to the other reviewers.
Round 3
Reviewer 2 Report
I thank the authors for the the improvements made throughout the manuscript.
The authors replied suitably to previous concerns with the exception of comment 12 (statistical FDR calculation). The authors’ response is not appropriate to what I tried to point out, yet I do recognise it is their manuscript and if they prefer to present a less robust analysis than they should, it is their choice.
I minor concern relating to the previous comment n. 6:
It must be stated in all the figure captions that the values presented are BC values relative to control.
Caption of Figure 4 should be corrected to “Intensity of phospholipid species containing 18:1 and 18:2 fatty acids relative to control in study 2”.
Caption of Figure 6 should state what is being compared. Intensity of lipid species in each BC stage relative to control in study 2?
I've made a few notes on the manucript text (see attached file).

Author Response
For Reviewer
Thank you for reviewing our manuscript (cancers-1256330). We appreciate your helpful suggestions and comments. We have revised the captions and sentences as you pointed out.
Your detailed review and comments have been very helpful in enabling us to revise our manuscript. We believe that our manuscript has been improved and strengthened by the changes made based on your comments.